# Niclosamide–Clay Intercalate Coated with Nonionic Polymer for Enhanced Bioavailability toward COVID-19 Treatment

**DOI:** 10.3390/polym13071044

**Published:** 2021-03-26

**Authors:** Seungjin Yu, Huiyan Piao, N. Sanoj Rejinold, Geunwoo Jin, Goeun Choi, Jin-Ho Choy

**Affiliations:** 1Department of Chemistry, College of Science and Technology, Dankook University, Cheonan 31116, Korea; 32182808@dankook.ac.kr; 2Intelligent Nanohybrid Materials Laboratory (INML), Institute of Tissue Regeneration Engineering (ITREN), Dankook University, Cheonan 31116, Korea; 12192032@dankook.ac.kr (H.P.); sanojrejinold@dankook.ac.kr (N.S.R.); 3R&D Center, CnPharm Co., Ltd., Seoul 03759, Korea; geunwoo.jin@cnpharm.co.kr; 4College of Science and Technology, Dankook University, Cheonan 31116, Korea; 5Department of Nanobiomedical Science and BK21 PLUS NBM Global Research Center for Regenerative Medicine, Dankook University, Cheonan 31116, Korea; 6Department of Pre-Medical Course, College of Medicine, Dankook University, Cheonan 31116, Korea; 7Tokyo Tech World Research Hub Initiative (WRHI), Institute of Innovative Research, Tokyo Institute of Technology, Yokohama 226-8503, Japan

**Keywords:** niclosamide, poorly-soluble drug, montmorillonite, Tween 60, drug delivery, solubility enhancement, bioavailability

## Abstract

Niclosamide (NIC), a conventional anthelmintic agent, is emerging as a repurposed drug for COVID-19 treatment. However, the clinical efficacy is very limited due to its low oral bioavailability resulting from its poor aqueous solubility. In the present study, a new hybrid drug delivery system made of NIC, montmorillonite (MMT), and Tween 60 is proposed to overcome this obstacle. At first, NIC molecules were immobilized into the interlayer space of cationic clay, MMT, to form NIC–MMT hybrids, which could enhance the solubility of NIC, and then the polymer surfactant, Tween 60, was further coated on the external surface of NIC–MMT to improve the release rate and the solubility of NIC and eventually the bioavailability under gastrointestinal condition when orally administered. Finally, we have performed an in vivo pharmacokinetic study to compare the oral bioavailability of NIC for the Tween 60-coated NIC–MMT hybrid with Yomesan^®^, which is a commercially available NIC. Exceptionally, the Tween 60-coated NIC–MMT hybrid showed higher systemic exposure of NIC than Yomesan^®^. Therefore, the present NIC–MMT–Tween 60 hybrid can be a potent NIC drug formulation with enhanced solubility and bioavailability in vivo for treating Covid-19.

## 1. Introduction

COVID-19 has been affecting around 120 M individuals with 2.66 M deaths all around the world as of 16 March 2021. The ongoing pandemic is considered as the toughest one that the 21st century witnessed until now [1]. Therefore, there is an urgent need of medicines, although there are FDA-approved medications such as dexamethasone, which might be beneficial for patients in the critical condition [2], nafomastat [3,4], remdesivir [5,6] etc., and the latter was found ineffective with less or no complete recovery [7]. Therefore, the scientific community has been trying to make new medicines and vaccines to fight COVID-19. Since we have seen various but similar epidemics such as SARS and MERS, scientists have come to the conclusion that it is important to utilize the available FDA-approved drugs, which could be re-purposed for Covid-19 therapy [8,9,10]. This is mainly because the novel corona virus has a similar host entry mechanism as SARS, and both of them enter the host cells through angiotensin converting enzyme-2 (ACE-2) proteins [11].

Among the various FDA-approved drug candidates, the anthelminthic drug Niclosamide [12] (NIC; 5-Chloro-N-(2-chloro-4-nitrophenyl)-2-hydroxybenzamide) has been found to possess superior in vitro anti-viral effects [13], throwing light on the fact that it could be potentially useful for treating COVID-19. NIC could act through two major pharmacological pathways. (1) The first is blocking the endocytosis of SARS-CoV-2. It has been reported that SARS-CoV-2 can bind to ACE-2, by which it can infect the host cells [14,15]. However, the actual blocking mechanism of NIC on the ACE-2 protein is still not clear [16]. (2) NIC can inhibit the autophagy of SARS-CoV-2 through S-Phase kinase associated protein 2 (SKP2) blocking. A recent study showed that NIC can prevent SKP2 and enhanced autophagy, thereby reducing the MERS-CoV replication [17], and this might be the same mechanism for SARS-CoV-2, mainly because these viruses use the same ACE-2 protein for host entry. Scheme 1 shows the plausible mechanism of NIC inhibition on SARS-CoV-2.

However, the main limitation associated with NIC is its poor water solubility [18] and thereby limited bio-availability in many in vivo studies. In fact, improving the water solubility of poorly soluble drugs has been a long-existing challenge in pharmaceutics [19]. Accordingly, different strategies using water-soluble polymers [20], lipids [21], surfactants [22], and inorganic materials [23] have been employed. However, very limited publications and associated references are found in the literature, in which these excipients are combined as hybrids or composites for improved performances.

Therefore, to have an improved solubility and therapeutic advantage, it is necessary to protect them in a suitable carrier, which could effectively deliver the drug in a well-controlled manner. To do so, a wide range of nanoparticle-based drug delivery systems can be utilized for effectively combating viral diseases such as COVID-19 [24]. For example, core–shell Eudragit S100 nanofibers were developed through triaxial electrospinning to have a colon-specific sustained release of Aspirin, indicating that such modifications would be beneficial for treating viral diseases such as COVID-19 [25].

There have been various approaches to encapsulate NIC in a polymer matrix as an inclusion complex; for example, HP-β-CD has been used to complex with NIC, which showed controlled NIC release along with an improved pharmacokinetics (PK) [26]. However, the PK data showed no significant difference between control NIC and the NIC-inclusion complex [26]. Similarly, NIC was chemically modified with PEG (poly ethylene glycol) to improve its solubility; however, there were no PK data reported [27]. Therefore, it is important to have a suitable nanocarrier that could protect, maintain, and control the drug in a specific way, especially in vivo applications.

Inorganic carriers such as layered double hydroxides (LDHs) [28,29] and montmorillonite (MMT) [30,31,32,33,34] clays have been used for various biomedical applications [35], thanks to their excellent bio-compatibility, easy modification, and encapsulation capability for various therapeutic agents [36]. In particular, MMT has been approved by the FDA as a diluting agent for oral administration [36,37]. Such nanoclay-based materials have been vigorously studied for its potential use as a drug-carrier toward a wide variety of pathological conditions such as cancer [38], bacterial disease [39], bone regenerative medicine [40], wound-healing applications [41], reinforcing fillers for dental adhesives [42], etc. Structurally, MMT has two tetrahedral sheets, covering one octahedral sheet in between. The tetrahedral sub-units have oxide anions at the tip, which are oriented toward silicone atoms. Silicone can be frequently substituted by aluminum cations. On the other hand, the octahedral subunits contain aluminum ions that are substituted by magnesium or iron ions, surrounded by hydroxyl groups present at the axial end of octahedrons [43]. The surface charge of MMT, [Na_0.7_K_0.02_Ca_0.04_(Si_7.78_Al_0.22_)(Al_3.20_Mg_0.64_Fe_0.16_)O_20_(OH)_4_], becomes slightly negative. This is mainly due to the fact that the oxide anion is the dominant charge balancing one (Si^+4^, Al^+3^, Fe^+2^, Fe^+3^, Mg^+2^) in the layer, making a negative layer charge on the MMT surfaces [44], which is eventually compensated by the solvated interlayer cations (Na^+1^, K^+1^, Ca^+2^) in order to satisfy the charge neutralization condition.

Previously, our own research group has published several papers on MMT for drug delivery applications [45,46,47]. Here, we focused on MMT-based oral formulation to be utilized as a potential therapeutic carrier for NIC due to the above-mentioned reasons. In addition, the bio-inspired layered MMT has several advantages for oral drug delivery applications. For example, its high mucoadhesive nature enables the payloads to evade the harsh GI barrier [48]. MMT can act as a detoxifier in the intestine by adsorbing dietary, bacterial, and metabolic toxins as well as abnormally increased hydrogen ions observed in acidosis [49]. All these properties make MMT an excellent orally administrable carrier for NIC. Even though there have been various studies using MMT-based composite materials for drug delivery applications [50], our final goal is to improve the NIC bioavailability, which would further enhance its anti-viral therapeutic efficacy in vivo. For example, MMT/gelatin composite NPs (719–754 nm) were reported for encapsulating isoniazid, a tuberculosis drug, for its pH-dependent controlled drug release [51]. Similarly, PLGA/MMT composites were reported to have sustained paclitaxel (PTX) delivery to human colon cancer cells in vitro [48]. It is worth noting that MMT has been well utilized to make composite with polymers; however, there have been no attempts to make a suitable formulation to improve the solubility of NIC, except for a few reports, where hydroxy propyl beta cyclodextrin (HPβCD) [26], and chitosan [52] were used to encapsulate NIC. Compared to the existing MMT composites, the main goal on which we focused here is to achieve an effective NIC formulation that can be administered orally to have therapeutic plasma concentration to achieve anti-viral effects.

MMT, as reported before, possesses high bio-compatibility and sorption property [50], which allows it a suitable drug delivery agent. There are different ways for making suitable drug-loaded MMT clays. The major mechanism involved in drug loading/encapsulation with MMT is via hydrogen-bonding [53], ion–dipole interaction [54], and cation exchange [55].

In our case, the main research goal of the current study was to improve the solubility, thereby having a better PK pattern for NIC by encapsulating it in mucoadhesive inorganic materials such as MMT. Additionally, Tween 60 was physically coated in order to further improve solubility of NIC. Tween 60 is a polyoxyethylene (20) sorbitan monostearate. Here, the number 20 following the ‘polyoxyethylene’ part represents the total number of oxyethylene -(CH_2_CH_2_O)- groups found in the Tween 60 molecule, which has a large hydrophilic head group [56]. The main reason for selectively choosing Tween 60 was due to its high dissolution property compared to Tween 20, but it was almost similar as Tween 80 [57]. We believe that this would be the first study to encapsulate NIC into a clay material/Tween 60 for enhancing bioavailability toward COVID-19 therapy.

The main research questions we address here are as follows: (1) How can NIC be incorporated in the MMT? (2) How does Tween 60 incorporation in the final NIC/MMT affect the solubility of NIC? (3) What would be the in vivo fate of NIC in an animal model? The research hypotheses are given as follows: (1) NIC can be incorporated into the MMT through ion–dipole interaction (Scheme 2a), (2) Tween 60 can be attached on the final NIC/MMT through physical adsorption improving the overall solubility of NIC (Scheme 2b), and (3) NIC would be sustained in the blood plasma for a longer time mainly due to the controlled release associated with MMT and would maintain the therapeutic window for the desired time.

Therefore, our current study mainly highlights the synthesis of NIC–MMT and Tween 60-coated NIC–MMT hybrids, and their in vitro drug release studies under various conditions (such as gastric and intestinal pHs), including in vitro analysis by Fourier transform infrared (FT-IR), XRD, and detailed PK data, which will be compared with commercially available Yomesan^®^.

## 2. Results and Discussion

### 2.1. Powder X-ray Diffraction (PXRD) Analysis

According to the synthetic method (Section 3.2) of the NIC–MMT hybrids, hereafter, the NIC–MMT hybrid samples were abbreviated as follows: NIC(1.0)–MMT, NIC(1.0)–MMT-E, and NIC(1.3)–MMT-E. Figure 1 shows the powder X-ray diffraction (PXRD) patterns of intact NIC, pristine MMT, NIC(1.0)–MMT, NIC(1.0)–MMT-E, and NIC(1.3)–MMT-E hybrids. The characteristic peak of (*001*) for pristine MMT was seen at 8.9° (Figure 1b), which was shifted to 6.9° for NIC(1.0)–MMT, NIC(1.0–MMT-E, and NIC(1.3)–MMT-E hybrids (Figure 1c–e). According to the Bragg’s law (nλ = 2d sin θ; λ = the wavelength of incident wave, d = the spacing between the planes in the atomic lattice, and θ = the angle between the incident ray and the scattering planes) [35], the basal spacing (*d*) increased from 9.9 to 12.8 Å for NIC(1.0)–MMT, NIC(1.0)–MMT-E, and NIC(1.3)–MMT-E hybrids, indicating the successful intercalation of NIC into the interlayer space of MMT. MMT has an expanding lattice in the *c* axis. To analyze the physico-chemical properties of 2-dimesional materials and their intercalates, PXRD is frequently used for solving their crystal structures, because they show, in general, well-developed (*00l*) diffraction peaks corresponding to the *d*-value that is increased to larger values upon intercalation [58]. XRD peaks of intact NIC crystalline were observed after hybridization, suggesting that the NIC molecules were distributed both in the MMT interlayer space and on the MMT surface.

### 2.2. Fourier Transform Infrared (FT-IR) Analysis

Figure 2 shows the Fourier transform infrared (FT-IR) spectra of intact NIC, NIC(1.0)–MMT, NIC(1.0)–MMT-E, and NIC(1.3)–MMT-E hybrids. The intact NIC showed major characteristic peaks at 3577 cm^−1^, 3490 cm^−1^, 1650 cm^−1^, 1517 cm^−1^, and 570 cm^−1^, those which could be assigned as -OH, -NH, -C=O, -NO_2_, and C–Cl groups, respectively (Figure 2a) [59,60]. The pristine MMT showed a characteristic broad band near 3400 cm^−1^ due to the –OH stretching vibrations of interlayered water molecules (Figure 2b). The band at 3630 cm^−1^ was corresponding to the –OH groups originating from the Al–OH, and that at 1041 cm^−1^ was from the Si–O bonds bound to the aluminum octahedrons in MMT [61]. The characteristic band at 915 cm^−1^ was due to Al–Al–OH stretching, and the band at 530 cm^−1^ was due to the Al–O–Si stretching [29,35]. After the hybridization (Figure 2c–e), all characteristics peaks of NIC and pristine MMT were overlapped within those of NIC-loaded MMT hybrids.

### 2.3. Determination of NIC Content

The NIC content of NIC(1.0)–MMT, NIC(1.0)–MMT-E, and NIC(1.3)–MMT-E were determined to be 25.7 ± 1.2%, 27.2 ± 0.8%, and 32.8 ± 0.7%, respectively, which was in good agreement with the theoretical content of 27.3%, 27.3%, and 32.8%, respectively. For Tween 60-coated NIC(1.0)–MMT, NIC(1.0)–MMT-E, and NIC(1.3)–MMT-E, the NIC content was reduced down to 15.5 ± 0.1%, 15.9 ± 0.6%, and 17.7 ± 0.4%, respectively, due to the incorporation of Tween 60 through the coating process employed in this study (NIC–MMT hybrids:Tween 60 = 3:2 *w*/*w*).

### 2.4. In Vitro Release Study

We pursued to enhance the bioavailability of NIC after its oral administration. We attempted to employ an advanced method for enhancing the solubility of insoluble drug based on the intercalation chemistry. In general, clay such as MMT has been utilized as a hydrophilic drug delivery carrier, since drug molecules could be encapsulated and stabilized in the interlayer space by intercalation reaction [62,63,64].

To check the in vitro release behavior (Figure 3, Appendix A) of NIC molecules out of a 2D lattice of MMT, the drug release profiles of intact NIC, NIC(1.0)–MMT, NIC(1.0)–MMT-E, and NIC(1.3)–MMT-E hybrids were made under a gastric condition at pH 1.2. According to the NIC release behavior, both NIC(1.0)–MMT and NIC(1.0)–MMT-E retained a continuous dissolution pattern, whereas in the case of NIC(1.3)–MMT-E, it showed a decline in its NIC dissolution. It should be noted that the latter has ≈30% excess NIC of cationic exchange capacity (CCE) for MMT than the former samples, meaning that excess NIC could be adhered on the surface in addition to their intercalated form. A sudden decrease in NIC dissolution from NIC(1.3)–MMT-E might be because under acidic conditions, the excess amount of NIC could be dissolved slowly but quickly aggregated and recrystallized. This peculiar mechanism could affect the dissolution pattern for NIC(1.3)–MMT-E, where it initially showed ≈55% NIC release compared with the other samples. A similar tendency was already reported recently by Jara et al. (2021) [60], showing that the hydrophobic NIC could be recrystallized under acidic conditions. As shown in Figure 3a, only 8% of NIC was dissolved and released from intact NIC even after 2 h under this gastric condition, owing to its poor aqueous solubility. However, different from intact NIC, the NIC release from NIC (1.0)–MMT, NIC(1.0)–MMT-E, and NIC(1.3)–MMT-E hybrids was significantly enhanced. Among them, the NIC(1.3)–MMT-E hybrid showed the highest drug release with 55.0% in 0.5 h. During the initial time points, the bursting effect was observed for the NIC(1.3)–MMT-E hybrid, due to the release of NIC molecules physically adsorbed on the external surface of the MMT carrier.

Under an intestinal condition of pH 6.8, as shown in Figure 3b, a slight enhancement of NIC dissolution was observed for intact NIC (15.0%) during the first 2 h due to an increased solubility at higher pH. On the other hand, for the NIC(1.3)–MMT-E hybrid, the release rate of NIC was determined to be very fast; more than 75% NIC was released during the first 10 min. This must be due to the rapid dissolution of NIC molecules presented on the external surface of the NIC(1.3)–MMT-E hybrid. On the other hand, NIC was released out in a sustained manner for up to 10 h for the NIC(1.0)–MMT and NIC(1.0)–MMT-E hybrids. In the case of the former, a fast release of NIC within the first 10 min (≈43%) could be explained by the partial deintercalation of NIC molecules, whereas for the latter, the release was found to be slower. Such a sustained release without any bursting effect, as shown in Figure 3, could be rationalized by the fact that NIC intercalated in the interlayer space of MMT was diffusively deintercalated out of a two-dimensional lattice. Therefore, it is concluded that the present hybrid drug, NIC(1.0)–MMT-E, can be an ideal formulation from the viewpoint of the controlled release performances under gastric and intestinal conditions.

### 2.5. Particle Size Analysis

The particle size distributions of pristine MMT, NIC–MMT hybrids, and Tween 60-coated NIC–MMT hybrids in deionized water were assessed by the dynamic light scattering (DLS) method. As shown in Figure 4, the average particle size of pristine MMT was determined to be 447 ± 41 nm in deionized water. The NIC(1.0)–MMT, NIC(1.0)–MMT-E, and NIC(1.3)–MMT-E hybrids have an average primary particle size of 550 ± 87, 448 ± 44, and 545 ± 48 nm, respectively. However, the particle size distributions of Tween 60-coated NIC(1.0)–MMT, Tween 60-coated NIC(1.0)–MMT-E, and Tween 60-coated NIC(1.3)–MMT-E were determined to be 656 ± 84, 697 ± 115, and 629 ± 102 nm, respectively, which were larger than that of uncoated NIC–MMT hybrids. These results indicate that NIC–MMT hybrids were well coated with Tween 60.

### 2.6. In Vivo Pharmacokinetics of NIC

In order to evaluate the effect of Tween 60 coating for NIC–MMT hybrids on the blood retention of NIC, Tween 60-coated NIC(1.0)–MMT, Tween 60-coated NIC(1.0)–MMT-E, and Tween 60-coated NIC(1.3)–MMT-E were orally treated to rats, and the mean plasma concentration of NIC was compared with Yomesan^®^, as a control. The mean plasma concentration–time profiles and the relevant pharmacokinetic parameters of orally administered Yomesan^®^ and Tween 60-coated NIC–MMT hybrids in rats are shown in Figure 5 and Table 1, respectively. Appreciably, higher NIC levels were measured in the rat plasma administered with Tween 60 coated NIC–MMT hybrids as compared with Yomesan^®^. According to the literature [65,66,67], the polymer surfactants such as Tween 20, Tween 60, Tween 80, Brij 35, Brij 72, etc. in formulation could result in an improvement in drug solubility and oral bioavailability. In particular, the highest NIC levels were found in the rat plasma administered with Tween 60-coated NIC(1.0)–MMT-E (Figure 5). As shown in Table 1, the parameters, C_max_, AUC_(last)_, and AUC_0-∞_ for NIC were 2.0-fold, 1.6-fold, and 1.5-fold higher for Tween 60-coated NIC(1.0)–MMT-E than for Yomesan^®^, respectively, with a considerable increase in T_max_. The increased systemic exposure of NIC became evident after oral administration of Tween 60-coated NIC(1.0)–MMT-E, suggesting that the sustained release of NIC out of MMT layers was efficiently made for enhancing the plasma NIC concentration and its circulation time. These results could also be explained by the mucoadhesive property of the present drug carrier, MMT [35,46,68,69], in which the hydroxyl groups of MMT lattice could form strong hydrogen bonding with the mucous layer in the rat intestine. Moreover, the controlled dissolution profile of NIC molecules from NIC(1.0)–MMT-E under the intestinal condition during 10 h suggested that the low solubility of NIC could be remarkably increased by the deintercalation of NIC from NIC(1.0)–MMT-E layers (Figure 3). Consequently, the controlled release property and the increased solubility of NIC from NIC(1.0)–MMT-E in gastric and intestinal fluids, respectively, could contribute to the enhanced oral bioavailability of poorly soluble drugs.

Thus, the in vivo results clearly indicated that our ternary hybrids system could potentially improve the bio-availability of NIC and could be extended for future anti-viral evaluation in vitro and in vivo. We hope that such clay-based systems could be engineered in a specific way in order to improve the dissolution kinetics of many poorly soluble but active drug molecules in the near future. Apart from the very successful in vivo PK data, we believe that ours is the first report on NIC intercalated into MMT clays for oral administrable COVID-19 formulation. Such clay-based intercalated active drug molecules would possess excellent mucoadhesion properties, which might be useful for developing various oral drug formulations toward COVID-19.

## 3. Materials and Methods

### 3.1. Materials

NIC and Yomesan^®^ was generously supplied by CnPharm Co., Ltd. (Seoul, Korea). MMT (Kunipia-F; CEC = 115 mequiv./100 g) and Tween 60 were purchased from Kunimine Industries (Tokyo, Japan) and TCI (Tokyo, Japan), respectively. The chemical formula of MMT is Na_0.7_K_0.02_Ca_0.04_(Si_7.78_Al_0.22_) (Al_3.20_Mg_0.64_Fe_0.16_)O_20_(OH)_4._

### 3.2. Preparation of NIC–MMT and Tween 60-Coated NIC–MMT

The synthesis of NIC–MMT hybrids was realized by the mechano-chemical method as follows; the mixture of sodium MMT and NIC powders was ground in a mortar for 1 hour without solvent for preparing NIC(1.0)–MMT, and with 26 mL EtOH (99.9%) for preparing NIC(1.0)–MMT-E and NIC(1.3)–MMT-E. The molar ratio between NIC and MMT was 1.0:1.0 and 1.3:1.0. The amount of added NIC was 1.15 mmol/1g of MMT. After grinding, the mixture was dried in vacuum for 16 h to obtain the NIC–MMT hybrids [70]. To prepare Tween 60-coated NIC–MMT hybrids, the suspension of NIC–MMT hybrids was prepared in EtOH (99.9%), where Tween 60 was dissolved (NIC–MMT hybrids:Tween 60 = 3:2 *w*/*w*). The suspension was again dried by rotary evaporator to get powder samples under the following conditions: temperature of a water bath = 50 °C; rotary evaporator speed = 100 rpm; and degree of vacuum pressure = 95 mbar.

### 3.3. Characterization of NIC–MMT Hybrid

The PXRD pattern for the NIC–MMT hybrid was obtained with a Bruker D2 Phase diffractometer (Karlsruhe, Germany) equipped with Cu K_α_ radiation (λ = 1.5418 Å). All the data were recorded at the tube voltage and the current of 30 kV and 10 mA, respectively. The FT-IR spectra were recorded with a Jasco FT/IR-6100 spectrometer (Japan) by the standard KBr disk method in transmission mode (spectral range 4000–400 cm^−1^, resolution 1 cm^−1^, 40 scans per spectrum). The particle sizes of the pristine MMT, NIC–MMT hybrids, and Tween 60-coated NIC–MMT hybrids were measured by using a particle size analyzer (ELSZ-2000ZS; Otsuka, Japan) in deionized water as the solvent. All the measurements were done in triplicates (n = 3).

### 3.4. Determination of NIC Content

To determine the NIC content in NIC–MMT hybrids and Tween 60-coated NIC–MMT ones. Each sample (10 mg of NIC–MMT hybrids) was dispersed in EtOH (99.9%) and sonicated for 30 min to extract NIC completely from NIC–MMT hybrids and Tween 60-coated NIC–MMT ones, respectively. The suspension was filtered through a polyvinylidene fluoride (PVDF) membrane with 0.45 μm pore (USA), which was then measured at 333 nm by a Jasco UV/Vis spectrometer (V-630, Easton, MA, USA).

### 3.5. In Vitro Drug Release Experiment

The in vitro drug release experiment was performed according to the paddle stirring method with a DST-810 dissolution tester (Labfine, Seoul, Korea) [29,46]. The impeller was set at 50 rpm, and the bath temperature was maintained at 37 °C. Each sample with an equivalent amount of 15 mg NIC was dispersed in a release media, and an aliquot was collected at scheduled intervals. The sampled aliquot was filtered through 0.45 μm PVDF membrane filter (USA) and assayed with the UV/Vis spectrometer as described in Section 3.4. Each sample was measured in triplicate. In this work, two different experimental conditions were employed to mimic the fate of orally administered drug formulation. Thus, to simulate the biological fluid and residence time under gastrointestinal condition, the pH values of the release media were adjusted to 1.2 and 6.8 with 2% Tween 60, respectively [65], and the experiments were performed for 2 h and 10 h, respectively.

### 3.6. In Vivo Pharmacokinetic Study

Male Sprague–Dawley rats, weighing around 300 g, were allowed for free access to a normal chow diet and water for at least one week before the experiments. The rats were housed in a light controlled room, where the temperature and relative humidity were maintained at 23 ± 3 °C and 50 ± 5%, respectively. The experimental protocol was approved by the Institutional Animal Care and Use Committee (IACUC No. 20-KE-608) at the KNOTUS Co., Ltd. (Incheon, Korea). The rats were fasted overnight with free access to water before oral administration of Yomesan^®^ or Tween 60-coated NIC–MMT at a dose of 50 mg NIC per kg. For blood sampling, approximately 0.25 mL of blood was collected from each of the rats at 0, 0.25, 0.5, 1, 2, 4, 6, 8, 12, and 24 h after oral administration of Yomesan^®^ or Tween 60-coated NIC–MMT hybrids (Yomesan^®^: n = 5; or Tween 60-coated NIC–MMT hybrids group (1) NIN(1.0)–MMT: n = 5; (2) NIN(1.0)–MMT-E: n = 5; and (3) NIN(1.3)–MMT-E: n = 5). The collected blood samples were centrifuged at 13,000 rpm for 15 min and about 150 μL aliquot was collected from each of the blood samples. The sampled aliquots were then stored at −20 °C until analysis. Then, blood samples were analyzed by the HPLC- MS/MS by the L2 SCIENCE Co., Ltd. (Kyeonggi-do, Korea).

## 4. Conclusions

Even though NIC has been reported to possess superior in vitro anti-viral effects and could be potentially useful for treating COVID-19, its poor aqueous solubility and recurring problems such as low bioavailability are still challenging. For these reasons, we have developed Tween 60-coated NIC–MMT hybrid systems based on the drug carrier MMT with mucoadhesive property. In this study, we were successful in intercalating NIC molecules into a 2D lattice of MMT via the mechano-chemical method and deintercalating them out of MMT in a controlled manner. After formulating the NIC(1.0)–MMT-E hybrid with Tween 60, the solubility of NIC and the prolonged retention of the hybrid drug could be determined under gastrointestinal condition. Overall, we found that the present Tween 60-coated NIC(1.0)–MMT-E hybrid significantly improved the oral bioavailability of NIC in the PK studies, which was more than 1.6-fold higher than the control Yomesan^®^, a commercially available NIC. Therefore, it is concluded that the Tween 60-coated NIC(1.0)–MMT-E hybrid can be suggested as a promising hybrid drug delivery system to enhance the bioavailability of NIC.

## Data Availability

The data presented in this research study are available in this article.

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
