# Peer review of "Niclosamide–Clay Intercalate Coated with Nonionic Polymer for Enhanced Bioavailability toward COVID-19 Treatment"

_polymers, 2021, doi:10.3390/polym13071044_

Round 1

Reviewer 1 Report

The manuscript reports a hybrid composed of polymeric surfactant Tween 60, inorganic material MMT for promoting the dissolution and effective delivery of niclosamide, a poorly-water soluble drug. The contents are interesting and fall well within the scope of POLYMERS. I recommend its acceptance for publication after the following issues are well addressed.  

Most recent developments about treating COVID should be included in the INTRODUCTION section, such as Polymers 2020, 12(9), 2034 ; Current Drug Delivery DOI:10.2174/1567201817666201006144008 AND DOI:10.2174/1567201817666200916090710

  • A full background about the methods treating poorly water-soluble drugs should be concluded into several sentences to project your work’s merits, such as poorly water-soluble is a long-existing challenge in pharmaceutics ( Drug Deliv. 2021, 18 (1) , 2–3), hydrophilic polymers (Polymers, 2020, 12, 2421), lipids (Colloids and Surface B - Biointerfaces, 2021, 201, 111629), surfactants (POWDER TECHNOLOGY,2020, 375, 302-309); and also inorganic materials (Polymers, 2020, 12, 1679) are frequently exploited to provide alternative resolutions to this issue. However, very limited publications can be found, in which these excipients are combined as hybrids or composites for an improved performances. Your job is an excellent example on this point.        

  • Page7 line 216-217: “Such clay based intercalated active drug molecules would possess excellent mucoadhesion properties…”

  • Please give the rat a detailed description, such as weight, breed and so on, provide Animal Ethics certificate numberin your article.

Author Response

Please find the file as attached.

Reviewer 2 Report

In the manuscript, Choi and Choy et al. developed a drug delivery system with montmorillonite and non-ionic surfactant Tween 60 to improve the oral bioavailability of niclosamide. Generally, the experimental results support most of the authors’ claims. However, using similar drug delivery system has already been applied in many studies and the work is not very novel.

Detailed comments:

  1. Using montmorillonite and non-ionic surfactants, including Tween, as drug delivery systems has already been published in many papers (J. Drug Deliv. Sci. Technol. 2017, 39, 200-209, J. Mater. Sci. 2015, 50, 7303–7313, Biomaterials 2005, 26, 6068-6076, etc.). Why is this work unique/significant? What kind of improvements have been developed for this system?
  2. More background information regarding the montmorillonite drug delivery system should be given, including a short summary of its mechanism of encapsulation. It should also be mentioned that why authors chose Tween 60, instead of other surfactants.
  3. Page 2, Line 92. The authors claimed that NIC can be incorporated in to the MMT through ion dipole interaction. How did authors obtain this conclusion? No more explanation has been found through the manuscript.
  4. Figure 3a, the cumulative release for NIC(1.3)-MMT-E decreases from 0.5 to 2 h. However, from kinetic view, the released drug should not be absorbed again, therefore the cumulative release should always increase (or at lease keep at the same level) in all time range. How to explain this observed decrease?
  5. Scheme 1, please use a formal chemical structure of niclosamide with proper bond length.

Author Response

Please find the file as attached.

Reviewer 3 Report

The paper describes the obtaining of a new hybrid drug delivery system made of NIC, montmorillonite (MMT), and Tween 60, to overcome low oral bioavailability of NIC.

The results are proper for publishing in this journal, however, there are issues that must be solved before its recommendation for publication, such as:

  • Please, explain what 001 and 002 means in diffraction analyses section. Maybe a scheme of NIC/MMT intercalation-interraction would be helpful.
  • NIC-MMT samples obtained in ehanol were washed after the obtaining of MMT/NIC complex? Or only a mixture of MMT with NIC was obtained?
  • FTIR specta. The assignments for 400-600cm-1 should be provided, too. Are there any shifts to demonstrate stronger interractions of NIC with MMT? Have you tried to perform quantitative FTIR analyses on the obtained samples. I believe that more information could be extracted.
  • You affirmed that Tween 60 was coated on NIC-MMT. Which analyses demontrates that? Have you performed some analyses (e.g. DLS) to emphasize the size distribution by intensity for the obtained polymer coated MMT/NIC?
  • I think that the application of this hybrid in the treatment of covid 19 is a bit extreme. Until this is demonstrated to be efective on infected subjects, I think it would be better to use potential application.
  • Have you prepared and analyzed a blank Tween 60-NIC or uncoated MMT-NIC as blank samples? I think it should be specified also as blank samples and analyzed throughout the manuscript against the other samples.

Therefore I would suggest publication of the paper after the major revisions are taken into consideration.

With respect,

Author Response

Please find the file as attached.

Round 2

Reviewer 1 Report

The authors have improved the manuscript’s quality substantially and their responses are clear and convinced. I recommend its acceptance for publication in its present form.

Reviewer 2 Report

The authors have addressed the questions properly and revised the manuscript according, therefore, the manuscript is recommended for publication in the present form.

Reviewer 3 Report

Dear Authors,

The manuscript was revised and modified according to the reviewer suggestions.

The manuscript has been improved and can be accepted for publication.

With respect,